# A Machine Learning Based Model for Energy Usage Peak Prediction in Smart Farms

SaravanaKumar Venkatesan [1] , Jonghyun Lim [1], Hoon Ko [2] and Yongyun Cho [1,*]

1   Department of Information and Communications Engineering, Sunchon National University, Jeollanam-do, Suncheon-si 57922, Korea; skumarvsk1288@gmail.com (S.V.); sshb56@s.scnu.ac.kr (J.L.)
2   Research Institute for Computer and Information Communication (RICIC), Chungbuk National University, Chungdae-ro 1, Seowon-gu, Cheongju 28644, Korea; skoh21@cbnu.ac.kr
*   Correspondence: yycho@sunchon.ac.kr

**Abstract: Context:** Energy utilization is one of the most closely related factors affecting many areas of the smart farm, plant growth, crop production, device automation, and energy supply to the same degree. Recently, 4th industrial revolution technologies such as IoT, artificial intelligence, and big data have been widely used in smart farm environments to efficiently use energy and control smart farms' conditions. In particular, machine learning technologies with big data analysis are actively used as one of the most potent prediction methods supporting energy use in the smart farm. **Purpose:** This study proposes a machine learning-based prediction model for peak energy use by analyzing energy-related data collected from various environmental and growth devices in a smart paprika farm of the Jeonnam Agricultural Research and Extension Service in South Korea between 2019 and 2021. **Scientific method:** To find out the most optimized prediction model, comparative evaluation tests are performed using representative ML algorithms such as artificial neural network, support vector regression, random forest, K-nearest neighbors, extreme gradient boosting and gradient boosting machine, and time series algorithm ARIMA with binary classification for a different number of input features. **Validate:** This article can provide an effective and viable way for smart farm managers or greenhouse farmers who can better manage the problem of agricultural energy economically and environmentally. Therefore, we hope that the recommended ML method will help improve the smart farm's energy use or their energy policies in various fields related to agricultural energy. **Conclusion:** The seven performance metrics including R-squared, root mean squared error, and mean absolute error, are associated with these two algorithms. It is concluded that the RF-based model is more successful than in the pre-others diction accuracy of 92%. Therefore, the proposed model may be contributed to the development of various applications for environment energy usage in a smart farm, such as a notification service for energy usage peak time or an energy usage control for each device.

**Keywords:** peak energy usage; machine learning; IoT smart farm; agricultural big data

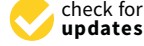

## 1. Introduction

The smart farm is an organization that helps to prevent crop damage. This smart farm has been collecting large datasets to enhance production predict energy usage. Smart farm goal is to maintain an automatic climate-energy control and improve growth conditions to increase the quality and quantity of the products [1]. It can produce these systems in a specific location without any restriction of farm conditions. Nevertheless, they must be designed according to the environmental conditions of the place where they will be installed. Control of the environmental energy is essential for the plant's growth because it represents ninety-five percentage of the yield of the crop, where the equipment, shape, and components of the smart farm depend on how much the outdoor climate differs from the needs [2]. From a smart farm perspective, greenhouse control is a complex

process. Recent advances in WSN have allowed us to express farming accurately to indoor climates. In addition, several control strategies are discussed in basic research studies to meet the desired requirements [3]. Machine learning and ARIMA methods have been proposed as an energy management strategy for smart farms [4]. For the energy management of smart farms, a step-by-step control approach is provided. A random multi-time energy management plan for greenhouses with energy sources was proposed. The data set integration problem of wireless sensor networks and the greenhouse monitoring system's hierarchical structure are explored [5]. This paper is a new integrated framework for estimating smart farm energy usage and crop production. Utilizing the machine learning and ARIMA models with integrated environmental climate-crop production models to develop energy activity systems supported a traditional control strategy, aim value of crop production, and factor value. It is proposed to use an IoT-based energy system to smart farm for crop growth [6].

The smart phase can play a more significant role in controlling stable energy supply in the smart farm. In addition, the integration of smart grid technologies that are new enables the smart farm to meet maximum standards regarding performance and environmental energy. With the progress of communication technologies, the automatic control of the indoor energy mate has found wide applications [7]. Although the initial objectives of growing in a smart farm environment were to provide a protected environment for the crop, the management of indoor energy has become a tool to guide crop production in the last five years [8]. A mathematical formula was developed for a group of microchips that proposes a strong distributed control strategy based on somewhat nest-built information [9]. It provided energy utilization in the smart farm of power micro grids. The purpose is to keep the amount of data stored in each micro grid around a value of reference by the control energy in the WSN of a micro grid [2]. We define accordingly the proposed model to a machine learning and ARIMA quadratic formulation [10]. Distributed control technology is proposed for interconnected smart farm residential utility group energy planning [11]. Previous approaches to machine learning models LR, XGB, SVM, RF, GBM, kNN, ARIMA, and ANN consider peak energy using indoor and outdoor climatic conditions and environmental interactions. Now, in addition, it focuses only on one regulatory aspect (climate control and smart farm energy management) [12]. In contrast, smart farm energy is related to indoor climate, interaction, energy consumption, and efficient control. In addition, this article focuses on analyzing the peak energy of the smart farm. At the same time, we have not yet explored the economic and environmental potential benefits that may arise from the integrated energy management of a team of smart farms.

The system integrates energy production from solar energy and its storage and data collection via a wireless sensor network [13]. This article proposes innovative concepts for applying the smart farm concept in the field of agriculture. Modeling, control, and optimization of machine learning and ARIMA model powered by the energy system are described. The model is viewed to improve smart farm economy, crop production, consistency, and efficiency. This method assumes that the sensor-derived energy can measure the properties related to data production, thus succeeding in non-destructive growth monitoring [14]. Factors enhanced statistics using machine learning LR, XGB, SVM, RF, GBM, kNN, and ANN models, i.e., different adaptive lag strips, are the best performance of the ML method [15]. Other processes such as computer mechanical IoT sensor-based rules for evaluating development-related properties have yielded encouraging results [16]. This article is about achieving the best calculations of smart farm properties for energy. The energy-smart farm is used to model the relationship between environmentally related developmental characteristics LR, XGB, SVM, RF, GBM, kNN, ANN and the time series ARIMA model. Following planned production, including smart machine learning to generate smart farm energy from large data environments and energy efficiency, this paper expects implementing machine learning and time-series ARIMA with precision energy sensors of similar characteristics to the smart farm.

## 2. Related Work

This research paper examines what has been done in the literature on the application of machine learning in environmental energy forecasting.

Nguyen, Xuan Hoai 2020 used statistical ML methods to develop data-driven forecasting systems that provide cost-effective solutions and improved performance. Meanwhile, the auto-regressive integrated moving average is a well-known linear statistical model for time series forecasting ARIMA. Prediction results for the RF model got the best performance [17]. Ashok Tatapudi, p Suresh Varma 2012 compared the performance of the machine learning algorithm with an increase in the upgrade energy approach for the study of IoT connections. As early smart farm environmental information, temperature and humidity are assessed to increase crop yield and evaluate crop losses [18]. Similarly, Ullah, I., Fayaz, M., Aman, M., & Kim, D. (2021) developed energy forecast IoT that can help improve the quality of life by transforming traditional technology into a smart farm. As the world's population grows, greenhouse gases become the best solution to global food demand. Greenhouses offer year-round production of fresh vegetables with a production rate of 50% higher than open-field cultivation. Fayaz, M., Aman, M., & Kim, D. (2021) proposed a machine learning algorithm plan to predict energy consumption and desired smart farm climate systems such as temperature, $CO_2$ levels, and humidity. Comparative analysis shows that the best performance obtained for a high crop growth outside is 26.56% and that for greenhouse is 27.76% [19]. Patokar, A. M., & Gohokar, V. V. (2018) investigated wireless transceiver hardware platform that wirelessly connects each node to the base station for data collection. Temperature, humidity, sprinkler water flow, and soil moisture data are then fed into the computer and displayed on the screen. The IoT can make decisions and control actions based on the collected data; they proposed infrastructure on intel's Galileo General-2 platform for low power to monitor, control, and support decision-making using the Internet of Things (IoT) [13].

The time series is made up of linear and nonlinear forms that are integrated. Many hybrid systems combining linear and nonlinear techniques have yielded relevant results in terms of accuracy compared to single models. However, because the best combination function for forecasting linear and nonlinear patterns is unknown, this modeling is still developing. Lately, D.S. Oliveira, J.F., and de Mattos Neto, P.S., 2019 proposed a hybrid system that seeks the functionality to combine predictions of ARIMA-based linear and non-linear models with machine learning models [10]. Escamilla-Garcia, A., Soto-Zarazúa, G.M., Toledano-Ayala, M., Rivas-Araiza, E. and Gastélum-Barrios, A., 2020 proposed the applications of artificial neural networks in greenhouse technology and present how this type of model can be adapted in the coming years with new technologies such as the Internet of Things and machine learning. Most of the analyzed works use the feed-forward framework, while repetitive and hybrid networks are somewhat exploited in the various functions of greenhouses [1]. In Lin, X., Sun, X., Manogaran, G., & Rawal, B. S. (2021), the author's reactive energy utilization technology was used as an efficient energy consumption system for the smart farm by integrating a smart farm energy system and evaluating its potential benefits. It is used to constantly monitor the control variables and predict the best wind power display on the smart farm by ensuring that the minimum voltage is reached in each situation, minimizing the total power loss [5].

Cifuentes, J., Marulanda, G., Bello, A., & Reneses, J. (2020) compared the performance of machine learning techniques to predict temperature based on input parameters such as relative humidity, solar radiation, rain and wind speed measurements, among others, 1 step above the regional level, compared to the traditional artificial neurological network. Globally, support vector machines are preferred based on an excellent compromise between simplicity and accuracy [20]. De Gelder, A., Dieleman, J.A., Bot, G.P.A. and Marcelis, L.F.M., (2012) made predictions that closed and semi-closed smart farms summarize climate, crop growth and development, and crop yield. The technical principles of a closed smart farm are described and the macro-and microclimates that result. Crop physiology and yield are examined about the typical growth conditions found in secure smart farms. Finally,

commercial growers' experiences are shared. Compared to an open smart farm in temperate climates, a closed smart farm can save 25–35 percent on fossil fuel-derived energy [21]. Maljkovic, D., & Basic, B. D. (2020) investigate energy systems in the European Union as a vehicle for reaching energy efficiency targets. For heat consumption, this paper finds the high accuracy prediction model of final consumers in the district hearing system based on the billing dataset. The author chooses ML algorithms for test protection accuracy; the RF model is a high prediction accuracy value of RMSE [22]. In this paper, results show that RF models effectively identify significant inputs related to energy usage for crop production and explain the relationship between energy usage and internal temperature change in smart farms, for example. Recently published work by Chen, R. C., Dewi, C., Huang, S. W., & Caraka, R. E. (2020) shows that ML techniques can aid in the accurate prediction of a dataset based on the package of input features, such as previous data classification-based model, in which the highest accuracy of the model is the best classifier. Practically, this paper adopts random forest to select the important feature in classification [23].

Motivated by the success of energy forecasting models, we propose understanding the relationship between smart farm agriculture output and various predictors. We also investigate different machine learning algorithms (ANN, kNN, SVM, XGB, GBM, RF, and time series ARIMA models) in predicting crop production and peak energy.

*Problem Statement*

To help the farmers to improve the quality of agricultural products and services by reducing investment costs by maintaining the best suitable crop according to the current or predicted environment energy.

Moreover, smart farm environmental variables (internal temperature, humidity, heating temperature) alert is given from historical data to take action. The forecast of the environmental energy events helps for suitable planning of farms using the ML algorithm approach.

## 3. Smart Farming in IoT

IoT is a revolutionary technology that reflects the future of communication and computing. This article is about implementing IoT-based smart farming energy in smart farm agriculture [7]. Crop management, resource management, lower-cost agriculture, enhanced quality and quantity, crop monitoring and field monitoring are all made possible by the IoT [12].

### 3.1. IoT in Agriculture

Figure 1, Smart farming is a topic that encompasses many applications of agriculture, agro-technology, and IoT technology in smart farm agriculture [2]. Using sensors, data collecting, wireless networks, cloud platforms, and data analysis have transformed smart farms and the agricultural industry. IoT farming allows you to get more visibility, control and information about your farm [24]. Better time management, enhanced decision-making, and practical resource utilization fertilizer and irrigation are some of these variables that allow you to safeguard healthier crops, larger yields, and resources. These variables result in less effort, increased revenue, and lower costs.

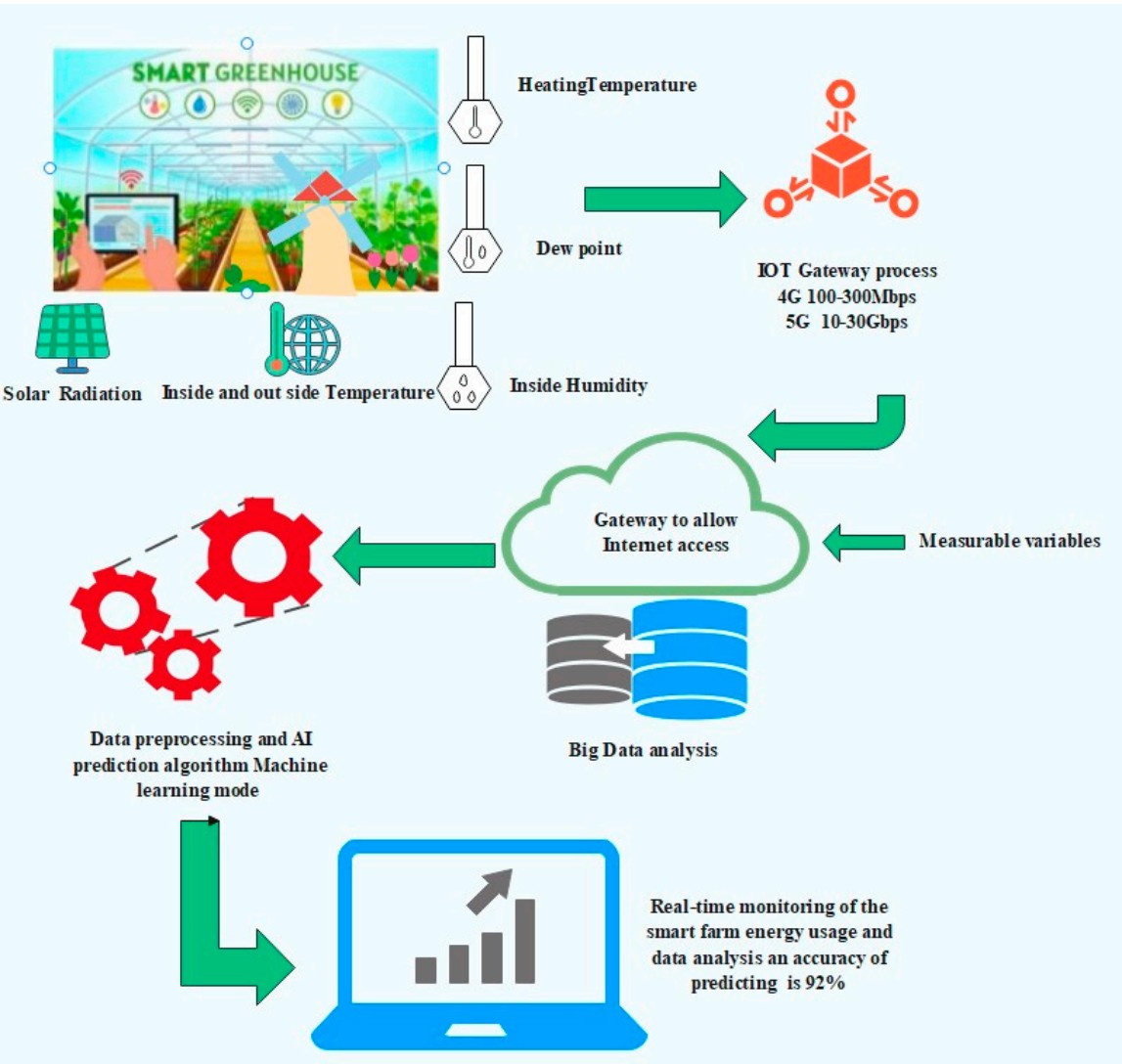

**Figure 1.** Smart farm steps up IoT agriculture.

### 3.2. Smart Farm Sensors

The IoT sensors used in the proposed model are a temperature sensor, heating temperature sensor, solar radiation sensor, dew point sensor, etc.

In this article, the dataset and all information are integrated into the Jeonnam Agricultural Research & Extension Service in South Korea for more information on new technologies in smart farming, the conventional farming methods used by the farmers, and the energy problem they face. A variety of sensors are used in IoT applications in connected agriculture to collect data in real time [25]. Farmers and agro-managers rely on connected sensors to evaluate soil conditions, monitor the health of crops and livestock, and adequately operate drones and agricultural equipment in distant places where there is not always access to Wi-Fi [26]. Farmers will have more options to use sensor technology in their particular settings for advancements in sensor technology, such as the integration of low-power, miniaturized and disposable tracking systems paired with the power of 5 G. They will be able to adjust to changing environmental, staffing, regulatory, and demand situations with the aid of this technology.

### 3.3. Wireless Sensor Network System

WSN makes it possible to communicate with the environment ubiquitously and seamlessly [27]. Temperature, light, motion, solar radiation, and humidity are all monitored

by the WSN's sensor devices, referred to as mods. The sensor points are scattered topically over a large open area. The network must be easy to deploy since the deployment environment may be harsh, hostile, and inaccessible [28]. WSN connects sensors to the internet using all available communication technologies. The IoT platform uses Wireless Sensor Networks to deliver efficient smart farm agricultural technology [29]. As illustrated in Figure 2, the integration of WSN in smart farm systems allows for real-time crop quality monitoring and optimization, as well as the possibility of large-area surveillance [30]. The continual monitoring of many environmental factors by dispersed sensor nodes across the paprika growth energy assists the farmer in supervising and maintaining ideal conditions for maximum yield while conserving significant amounts of energy.

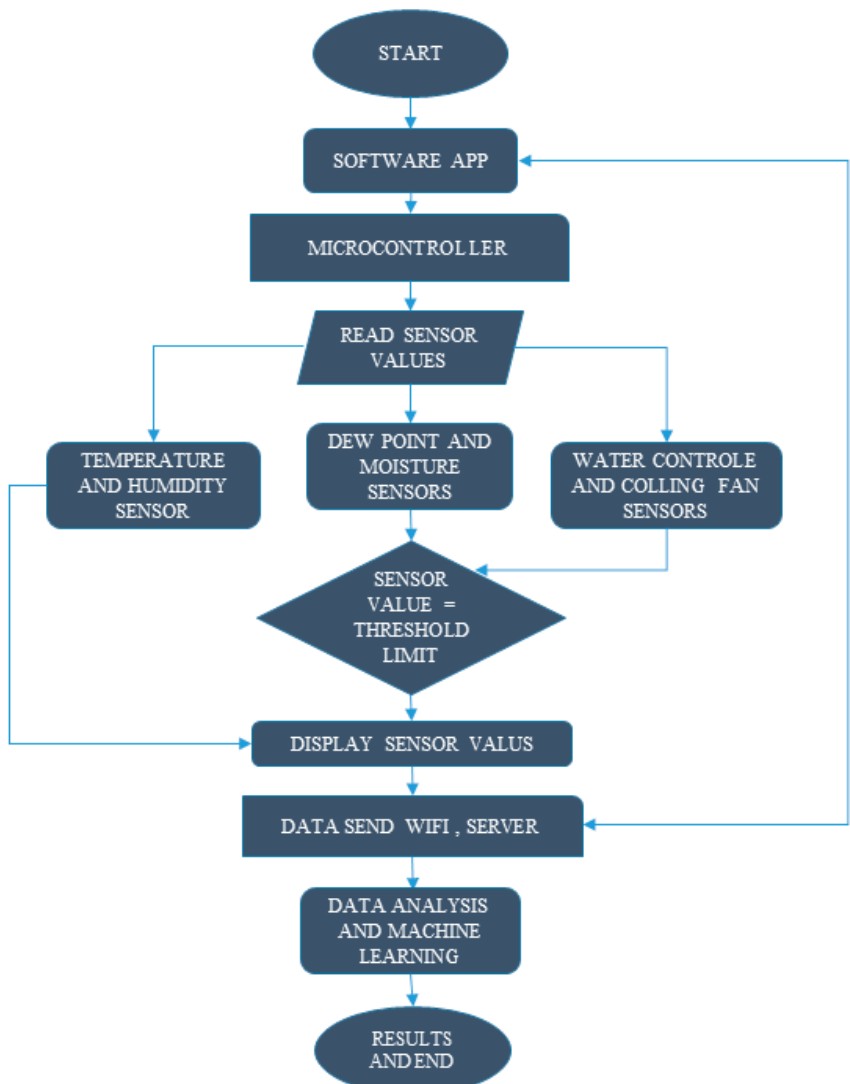

**Figure 2.** Smart farm sensor flowchart.

### 3.4. Smart Farming Energy

The restrictions of wireless sensor networks, i.e., Table 1, show that environmental energy description and the requests of the inside temperature, humidity, air heat temperature, outdoor temperature, solar radiation, dew point, and light monitoring collectively considered. The central concept is to investigate the gradual temperature fluctuations. The sensors, in particular, are designed to regulate their internal power levels based on the temperatures they observe. When the recorded temperature is average, the sensors will only wake up on rare occasions to find a peak energy efficiency. If the temperature is

abnormal, the sensor nodes lower their duty cycles to ensure that the temperature anomaly alert is active on time.

**Table 1.** Energy variable description.

| Data Variables | Description |
|---|---|
| Outside weather statuses (temperature, wind speed and solar radiation and humidity) | Inside the greenhouse is about 80 to 85 degrees Fahrenheit, wind speed continuous 1.2 = 2.50 mph |
| Indoor air temperature, humidity at different | RH mean daily mean relative humidity [50%] |
| Light inside the greenhouse | On average, greenhouses require six hours of direct or full-spectrum light per day. If natural illumination is not possible, additional illumination must be employed. Supplemental lighting uses a large number of high-intensity artificial lights to improve crop growth and productivity. |
| Soil humidity and air temperature | The relationship between soil moisture and near-surface air temperature is crucial for climate change and climatic extremes. Annual air temperature is inversely proportional to soil moisture, which results in dry wet soil warmer cold climates. |
| Dew point energy production | The humidity of 40% RH at 20 °C equals 6.0 °C dew point temperature. With a short dew point control band, it's easy to control the environment and save energy. With a short dew point control band, it's easy to maintain the environment and save energy |
| Heat exchange rates and power of energy pumps | Evaporator and condenser side temperatures, carrier fluid temperatures from various borehole heat exchangers, ground loop mass flow rate, and electrical power at the heat pump compressor and circulation pump were all monitored fluid temperatures from various borehole heat exchangers, ground loop mass flow rate, and electrical power at the heat pump compressor and circulation pump were all monitored. |

*3.5. Smart Farm Software Architected*

The smart farm system described in this study is based on the ability to regulate the culture environment of crops, which allows for faster plant selection for higher production, climate resilience, and regulation of agricultural information in real-time for larger producers [31]. Machine learning is also used for regulating the crop growing environment when it comes to adapting to climate change. Figure 3 shows the flowchart of an existing smart farm system. It is a structure that sends user comments and control values calculated by the server to the integrated controller to manage the growing environment in the installation [32]. The measured values are gathered via different sensors, sent to a cloud server, and displayed on a monitoring screen for users in the current architecture.

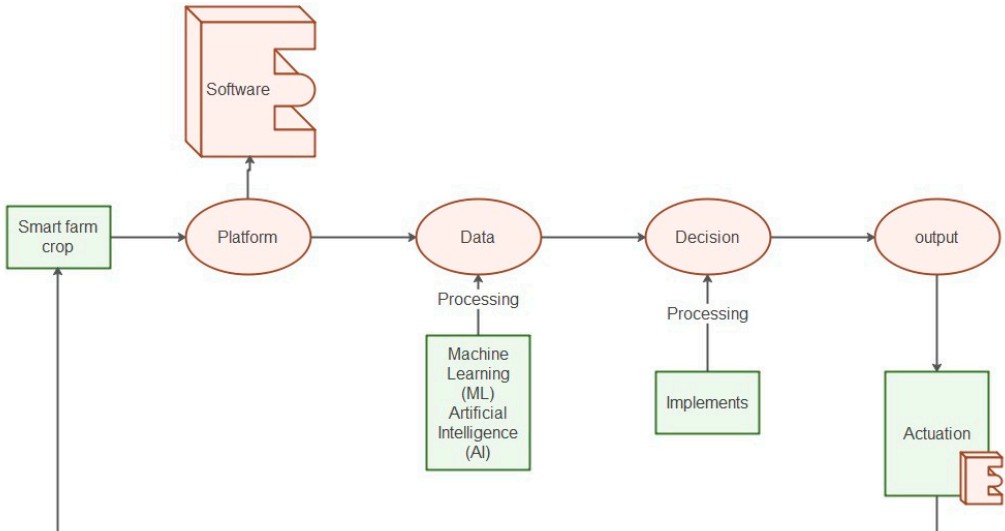

**Figure 3.** System architected in smart farm.

## 4. Methodology

### 4.1. Linear Regression

The most popular prediction model for determining the relationship between variables is linear regression. The idea is linear, regardless of whether the data are univariate or multivariate. Simple linear regression and multiple linear regression are two types of linear regression. The linear regression y is the independent variable, which can be continuous or categorical and x is the dependent variable, which is always continuous. It is investigated using probability distributions, emphasizing conditional probability distributions and multivariate analysis [33].

### 4.2. ARIMA Model

ARIMA is the most popular and efficient forecasting model. Time series data are data collected in sequence. The series might be yearly, monthly, weekly, hourly, or other time interval. The ARIMA model is a collection of auto-regressive, moving average, and auto-regressive moving average models. The differencing utilized to turn non-stationary data into stationary data ARIMA and kNN is the ARIMA model's integration component. Data that do not exhibit a trend are referred to as static data. ARIMA (p, d, q) is the acronym for ARIMA (p, d, q), where p, d, q denotes the data order [34]. The Partial Auto Correlation Function plot yields the value of p, whereas the auto correlation function (ACF) plot yields the value of q. The lag difference is used to calculate d.

### 4.3. Artificial Neural Network (ANN)

An artificial neural network is a machine learning method based on the notion of a human neuron. It is a computational model inspired by biology, having processing components such as neurons, and connections between them and coefficients weights connected to the connections. Because ANNs are inspired by brain structure, it is crucial to identify the critical components, dendrites, cell body, and axon function. Neurons are a network of fibers that carry electrical impulses from the cell body to the rest of the body. The signals are added and gathered by the cell body. The axon uses a lengthy fiber to transmit the signal from the cell body to the other neurons [35]. A synapse is formed when a cell's axon connects to a cell's dendrite. Neuronal networks' activities are thus determined by the arrangement of neurons and individual synaptic pressures. Figure 4 shows a broad diagram of a biological neuron, along with each component.

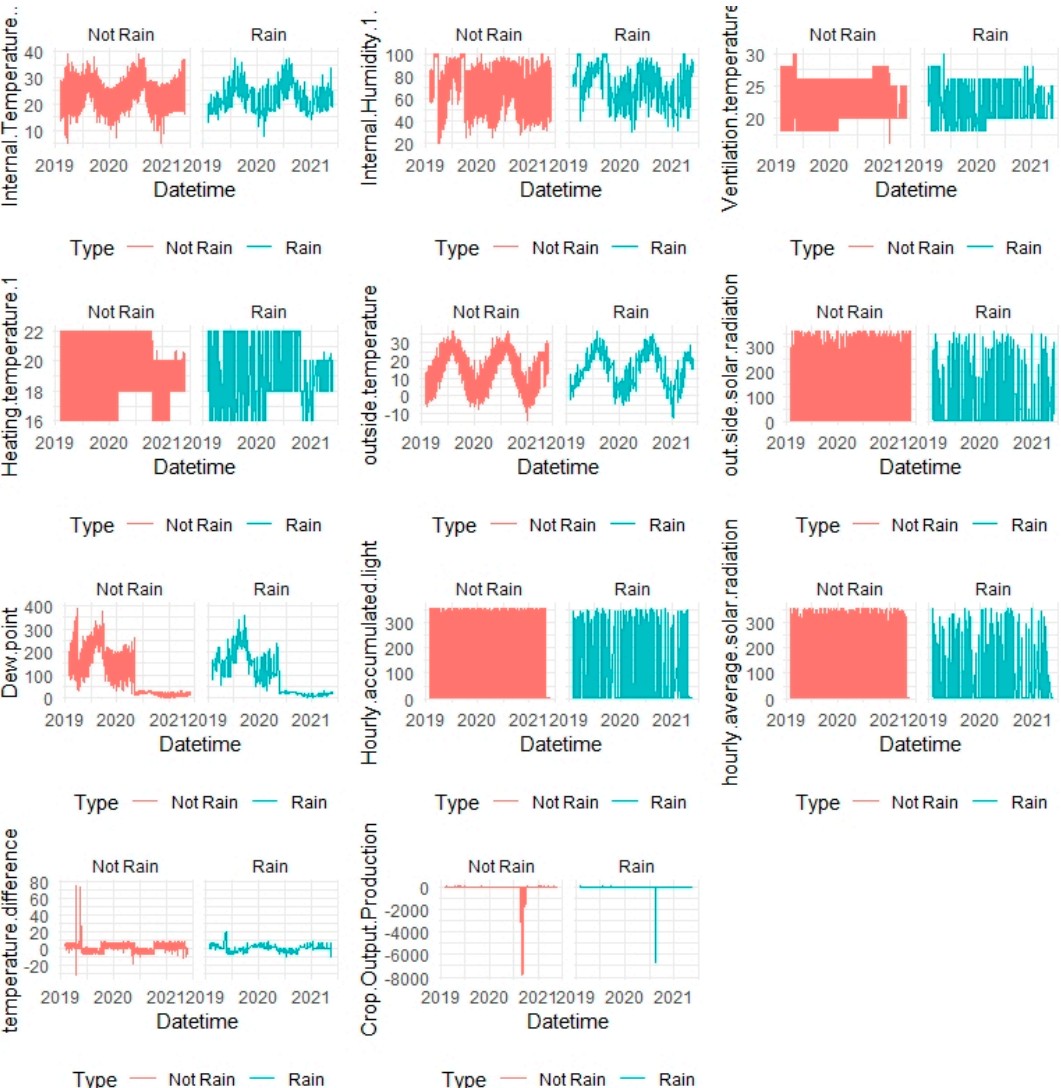

**Figure 4.** Energy dataset 2019 to 2021.

The average peak production rate across multiple runs and the peak rate generated over time are two metrics used to describe neuron activity. The speed at which a neuron creates these peaks is used to identify them in ANNs. Adaptive synaptic weights connect neurons to other neurons in the preceding layer, and knowledge is often kept as a series of weighted connections. A training procedure is carried out when these connection weights are adjusted orderly and using an appropriate learning technique. The machine learning technique presents the input to the network and the desired result, then changes the consequences so that the network can generate the desired output [35]. The weights will have important information after training, but it will be redundant and worthless before training.

The information processing starts with the inputs Xn, weighted and combined before passing through an activation function to create the output, represented as $\xi = \Sigma Xi \cdot Wi$ [34]. This activation value is multiplied by the particular weight (Wn) and transmitted to the next node for each outgoing connection. If a linear activation is taken into account, the result is $y = \alpha (wx + b)$.

### 4.4. k-Nearest Neighbor (kNN)

The metric used to compute distances between various instances significantly impacts k-nearest neighbor (kNN) classification accuracy. It was used to impute missing data in

this article [36]. In general, any model may be used for imputation; however, kNN was chosen for this investigation because it can produce satisfactory results while reducing computing costs to a minimum; it is widely used for this purpose. The model's premise is that a measurement's missing value is assigned to the most frequent value among the specified number of neighbors of that measurement [36].

### 4.5. Support Vector Machine (SVM)

The support vector machine is one of the most fundamental algorithms, and it is mainly used as a baseline in this study to evaluate the models' performance. The family of linear models lies at the heart of this approach. The model trains the original vector into a higher-dimensional space and searches the dividing hyperplane with the smallest gap. Two parallel hyperplanes are created on both sides of the hyperplane dividing classes. A separating hyperplane minimizes the distance between two parallel hyperplanes [37].

The technique is based on the idea that the larger the difference and distance between these parallel hyperplanes, the lower the classifier's average error. SVM stands for support vector machine learning, and it is a type of supervised learning commonly used for data grouping and anomaly identification. With a hyperplane, the SVM algorithm creates a model that optimizes the separation between data points in each category default is a line. Still, the model varies with the specified kernel. The SVM () function in R package e1071 can build a model given a training data set and predict the classification of additional data points [37,38]. SVM is helpful because it is fast and there is no danger of over-fitting the data.

### 4.6. Random Forest (RF)

It is a popular algorithm since it is simple to apply and easy to measure prediction components. It is based on the notion of a vast number of interconnected DTs. The random forest creates multiple trees and then combines them to produce accurate and reliable results [39]. The random forest algorithm aggregates the judgments of individual trees to enhance accuracy. To increase the software's accuracy, reuse the analytic technique, the random forest is used with the GBM [40]. Bagging is used to create unpredictability. Bagging stands for bootstrap aggregation, which improves the algorithm's accuracy and stability. As far as math is concerned, $X'$ is the prediction for the missing samples, b is the number of trees, b = 1,2,3 . . . B; and fb = Train a DT fb on Xb, Yb.

### 4.7. Gradient Boosting Machine (GBM)

The GBM learning method has grown famous for model prediction. GBM operates in three stages. The first step is to fine-tune the loss function, and the second step determines who is the weaker learner.

The third step is to lower the loss function, constructing by adding trees to the weaker learner and creating an adaptable model. The primary disadvantage of random forest is its slowness. So, to achieve better results for software reuse analytics, GBM is coupled with it to avoid performance issues. GBM is a method used in regression and classification procedures to improve prediction outcomes and reduce mistakes by minimizing bias [41].

### 4.8. eXtreme Gradient Boosting (XGB)

XGB technique is a new enhancement on the gradient boosting machine based on regression trees. The method is built on the concept of "boosting," which involves using additive training procedures to aggregate all the predictions of a group of "weak" learners to produce a "strong" learner. XGB's major objective is to reduce overfitting and underfitting issues while simultaneously reducing computational expenses through this prediction function [42].

## 5. Agricultural Big-Data

To produce valuable data that farmers can use, big data in the agriculture business relies on information, technology, and analytics. Big data may offer information for the agricultural industry or improve the efficiency of specific segments or places. It depends on big data mining processes to generate this critical information. Using the methodology basic shapes can be found in large datasets and this information can be compressed into digestible sections [43,44]. To cope with the growing demand for food and the pressures of climate change, policymakers and industry leaders are seeking the help of technological powers such as IoT, big data analysis, and cloud computing.

IoT devices aid the initial part of this process, data collecting. Sensors in smart farms and open agriculture and fields, optimum energy, solar radiation, soil, and plants help gather real-time data from the ground.

### 5.1. Dataset

This study shows the smart farm crop energy data from January 2019 to May 2021 (29 months) in Figure 4. They based these energy data on date time, internal temperature, internal humidity, outside temperature, ventilation temperature, heating temperature, outside temperature, outside solar radiation, dew point, hourly accumulated light, hourly solar radiation, temperature difference, paprika crop output production, etc.

The actual energy data are collected from smart paprika farms of the Jeonnam Agricultural Research and Extension Service in South Korea. The statistical collection for the smart farms paprika growth was most popular among the past researchers, including machine learning and time series ARIMA. Correlation is the first step in data analysis before moving on to more intensive research and selecting interrelated variables; next is to model the variable relationship using ANN, kNN, SVM, RF, GBM, and XGB and ARIMA. This study has given a full expansion of the site and smart farm agriculture, and it provides energy and environmental properties in the additional material and variable in Table 2.

**Table 2.** Material and variable.

| Data Variables | Type | Measurement |
|---|---|---|
| Internal temperature | Continuous | 18 °C to 26 °C |
| Internal humidity | Continuous | 60–80% |
| Ventilation temperature | Continuous | The temperature is 32 °C (90 °F) during the day and 24 °C (75 °F) at night. |
| Heating temperature | Continuous | 80 to 85 degrees Fahrenheit |
| Outside temperature | Continuous | Winter (−6~3 to 10 °C) |
| Outside solar temperature | Continuous | Spring (6 to 15 °C) |
| Dew Point | Continuous | Summer (20 to 32 °C) |
| Hourly accumulated light | Continuous | Fall (15 to 22 °C) |
| Hourly solar radiation | Continuous | 59 °F and 95 °F |
| Temperature difference | Continuous | Grams of water per cubic meter of air $(g/m^3)$ |
| Crop output production | Continuous | Moles of light (mol photons) per square meter $(m^{-2})$ per day $(d^{-1})$ |

### 5.2. Data Pre-Processing

Table 3, the model was trained using data from field deployments rather than a selected dataset. Pre-processing the existing data and obtaining meteorological data for the time frame chosen were two important steps in developing a prediction model. The statistics show that the area has many big data smart farms, with a maximum of 20,111 energy data in the environmental factors. Data more than three standard deviations from the mean value are omitted from all paprika smart farm energy usage environmental variables. Getting rid of those outlier increases prediction accuracy.

Missing data points—some power levels are not captured due to intermittent connectivity and system unavailability. Linear interpolation functions using these missing values supplied during pre-processing. Normalization—to eliminate outliers, the data normalized to $\pm[0,1]$.

*5.3. Dataset Correlation Analysis*

The exterior and internal variables collected from the computerized dataset, as indicated in the sensor data section count, mean, std, min to max (25% to 75%), and are shown in Table 3 with average $\pm$ constant deviation, constant error, and lowest and maximum values.

**Table 3.** Summary of descriptive statistics for each feature in the smart farm temperature data.

| Variable | Count | Mean | Std | Max |
|---|---|---|---|---|
| Internal temperature | 20,111 | 22.24 | 4.91 | 4.80 |
| Internal humidity | 20,111 | 70.39 | 16.79 | 19.10 |
| Ventilation temperature | 20,111 | 22.50 | 2.85 | 16.00 |
| Heating temperature | 20,111 | 19.22 | 1.99 | 16.00 |
| Outside temperature | 20,111 | 14.326 | 9.56 | $-14.20$ |
| Outside solar temperature | 20,111 | 75.35 | 98.01 | 1.00 |
| Dew Point | 20,111 | 91.67 | 84.83 | $-2.700$ |
| Hourly accumulated light | 20,111 | 86.129 | 116.42 | 1.000 |
| Hourly solar radiation | 20,111 | 73.475 | 96.65 | 1.000 |
| Temperature difference | 20,111 | 0.4061 | 4.0352 | $-32.70$ |
| Crop Output Production | 20,111 | $-347$ | 136.07 | $-78,962$ |

In this study, eleven different energy datasets (date-time, internal temperature, internal humidity, ventilation temperature, heating temperature, outside temperature, outside solar temperature, dew point, hourly accumulated light, hourly solar radiation, temperature difference, crop production) were used to assess the ML models. This has complicated recording each energy parameter because of the unavailability or uncertainty of sensor measurements. In the present study, Figure 5 follows the correlation coefficient approach to extract the best features, the most commonly observed method to examine the relationships between the properties. This type of correlation test may be used to determine whether or not the link between the independent and dependent variables is strong.

A regression plot in the smart farm is one useful way to depict the relationship between two features. Here we consider internal temperature vs. environmental variables. The best features of the correlation coefficient approach to extract the independent and dependent variables are the outside temperature, humidity, and heating.

Before performing regression analysis, it is important to verify correlations between feature variables. Strong integration can lead to unstable modeling results. Figure 6 shows the correlation coefficient of the smart farm energy rating matrix thermal map feature variables and label variables. R = 0 represents an uncorrelated connection; R = 0.4 represents a weak correlation; R = 0.4 to 0.75 represents a correlation; 0.75 |R| 1 represents a high correlation; R = 1 represents a completely correlated relationship, such as outside temperature with internal temperature, solar temperature and heating temperature, and ventilation temperature with heating temperature in Figure 6. Including various feature variables at the same time will affect modeling efficiency, causing feature selection first.

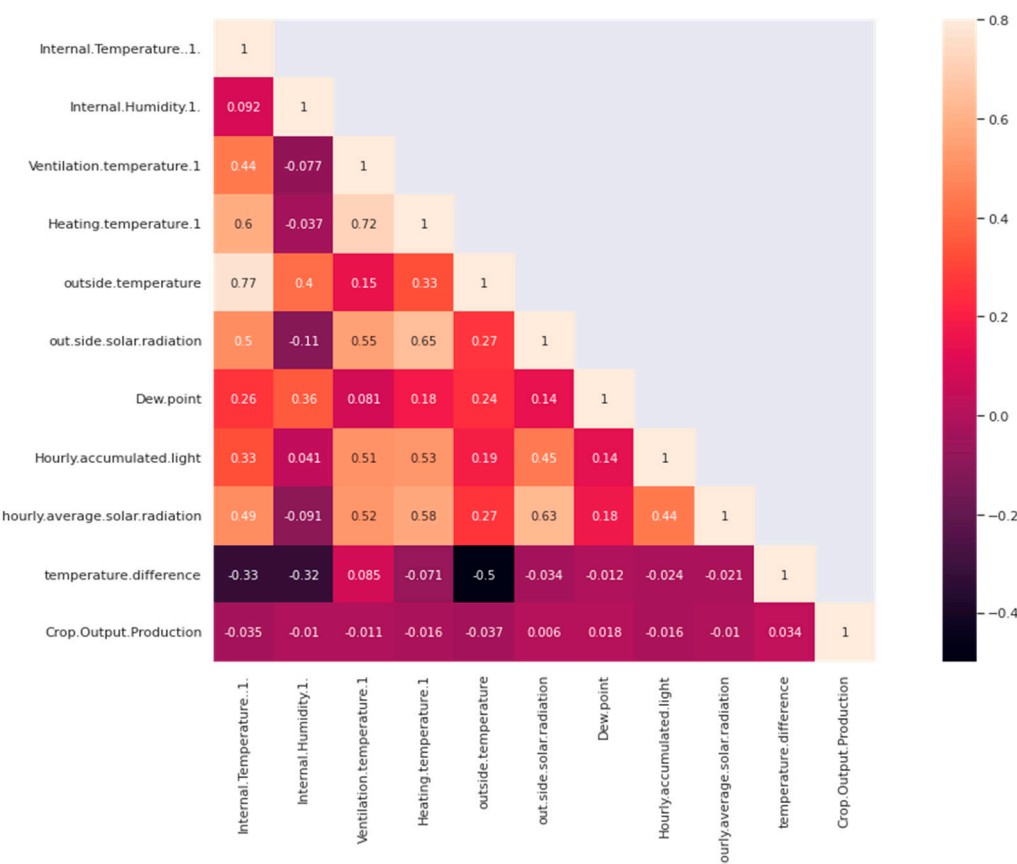

**Figure 5.** Correlation analysis result.

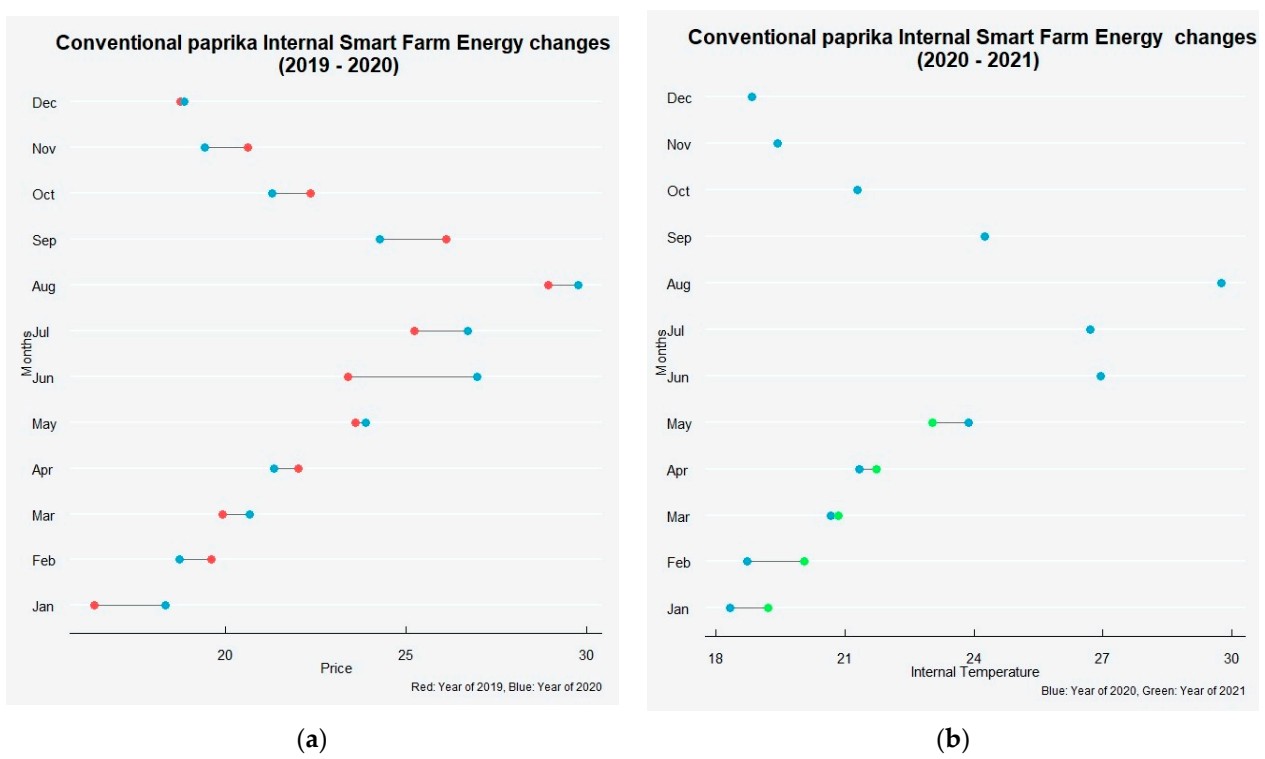

(**a**)                                   (**b**)

**Figure 6.** (**a**) Temperature change average result 2019 to 2020; (**b**) temperature change average result 2020 to 2021.

## 6. Data Analysis of Peak Energy Changes Monthly and Hour of the Day

The deployment of sensors on the smart farm's demand-side can collect environmental energy utilization data from greenhouse holders in real-time. In the time dimension, the energy utilization behavior of smart farms can be in an hour, a day, a month, or even a year. Therefore, greenhouse holders can describe the energy behavior of the environment in different time series from one hour to one year. The environmental energy use of a smart farm in an hour is very inconsistent. The environmental peak temperature application of greenhouse holds during the day often shows some differences at different times.

In contrast, many external factors usually affect the monthly peak energy changes behaviors. Figure 6 shows the yearly environmental peak energy variation of the usage of a smart farm in South Korea. The highest average energy of 30 °C is observed in August. The black cross represents outliers, which mostly belonged to the lower abutting. The lowest average energy, (2019) −10 °C and (2021) 10 °C, was observed from December to January. The most significant variation in energy was observed from July to September, in which June be attributed to seasonal weather fluctuations in this month of the year, respectively.

This paper shows the expected smart farm seasonal environmental temperature in Figure 7. The reference vector of temperature expressions of crops available on the smart farm and the average constant values, respectively, are equal to the seasonal temperature intervals each year in 2019 (winter 19.07 °C, summer 25.54 °C, spring 21.72 °C, fall 22.73 °C), 2020 (winter 18.65 °C, summer 27.64 °C, spring 22.02 °C, fall 21.55 °C), and 2021 (winter 19.5 °C and spring 21.98 °C) for Smart farm. Figure 8 shows that the expected environmental temperature paprika crop production will be, respectively, unequal at each hour of the day interval from 2019 to 2021. We should remember that a value of 0 [kW] shows that the crop receives enough sunshine. The expected ventilation temperature loads of the smart farm are supposed to be equal to 25.2 to 27.5. We note the local ensure the heating temperature usage defined by their efficiencies stated to be similar to 23 °C to 26 °C, which reactivated at the first time interval of each hour. The rest of the time should be "off." The humidity and dew point efficiency is set to be 75% to 110%. However, the set to be our variation humidity 05 a.m. to 7 p.m. [H] and dew point peak hour 8 a.m. to 7 p.m. should be equivalent. In the data analysis, solar radiation is very high energy compared with other environmental energy.

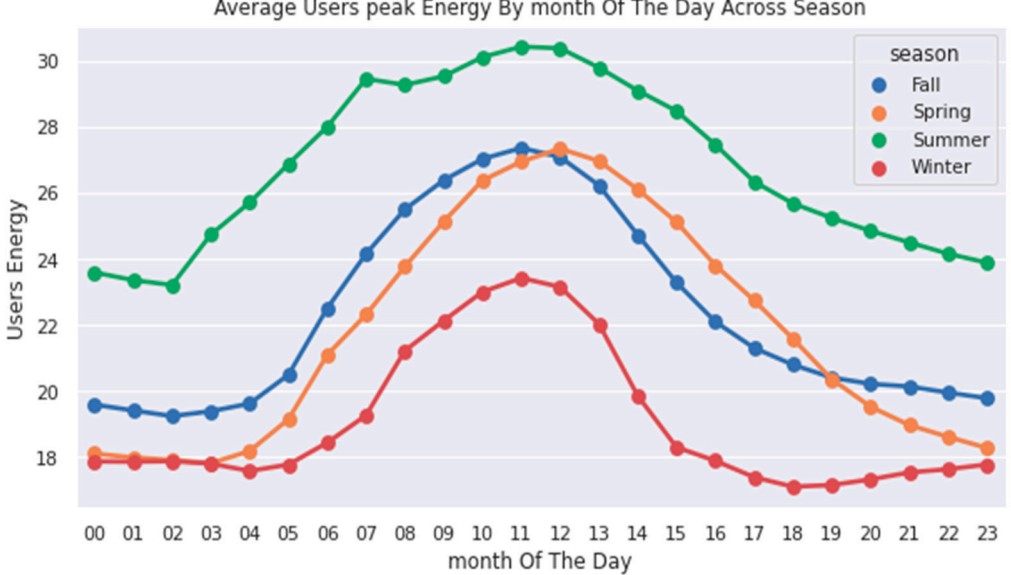

**Figure 7.** Seasonal temperature value 2019 to 2021.

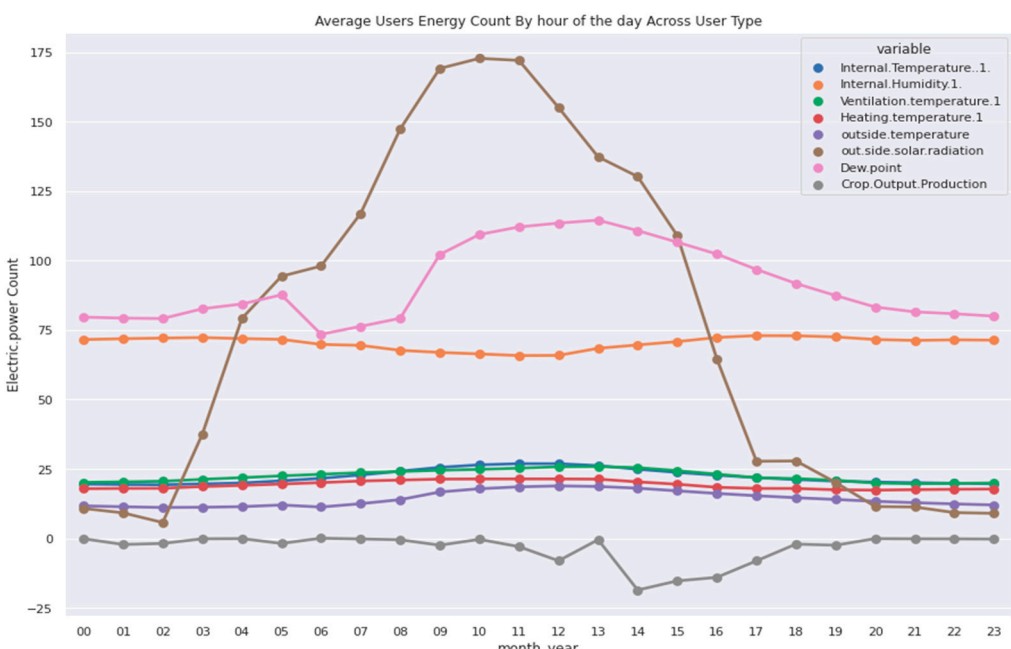

**Figure 8.** Peak energy temperature variable.

## 7. Model Performance

The research results revealed significant correlations between the chosen models and peak energy use. The foundation for identifying the best predictive model is that models with MAE, MSE, and RMSE values closer to zero are good predictive models, whereas R2 values are more comparable to the one that generated the best outcomes. RF is the best effective model for forecasting yearly energy use in this study. The best predictive model was an RF-based model, which provides strong results in big datasets, while the eighth best predictive model was an ANN-based model, which has gotten less attention in energy prediction. ARIMA, kNN, LR, ANN, GBM, XGB, SVM, and RF-based models are comparable to kNN stacking and LR. GBM splendidly outperforms kNN and XGB with better MAE, MSE, RMSE, and R-Squared. The Stacking and LR-based models perform the worst but take the least time to train.

The highest crop coefficient at the smart farm temperature is 0.45 and the lowest values or P-values of the critical coefficients are 0.05. An apparent correlation was found between the internal and outside temperature. The same observation is valid concerning the environmental condition parameters (ventilation temperature, heating temperature, outdoor temperature, outside solar temperature, dew point, hourly accumulated light, hourly solar radiation, temperature difference, and crop output production). These are strongly correlated with low coefficients of importance. The linear regression procedure is a method for assessing the influence of all predictors on the smart form in a selected manner. The analysis confirmed that the outdoor temperature, heating temperature, and humidity are the most important parameters in the emitted smart farm.

Hidden layer nodes were used to confirm the data calculated from the research, allowing ANN models to detect the link between smart farm input-output temperature energy variables. Figure 9 shows the $R^2$ and RMSE values for each ANN model using three distinct transfer mathematical functions for the training and testing stages. Many researchers have utilized these models to forecast the performance of smart farm crops.

The correlations between input variables and temperature may be nonlinear in this study due to the higher efficacy of nonlinear functions. The performance of ANN models with a linear function like purlin was lowered due to high RMSE values in the training and testing stages, as indicated in the findings. The drop in efficiency is most likely due to the function, which performs an essential linear modification on the processed input before delivering it to the output layer.

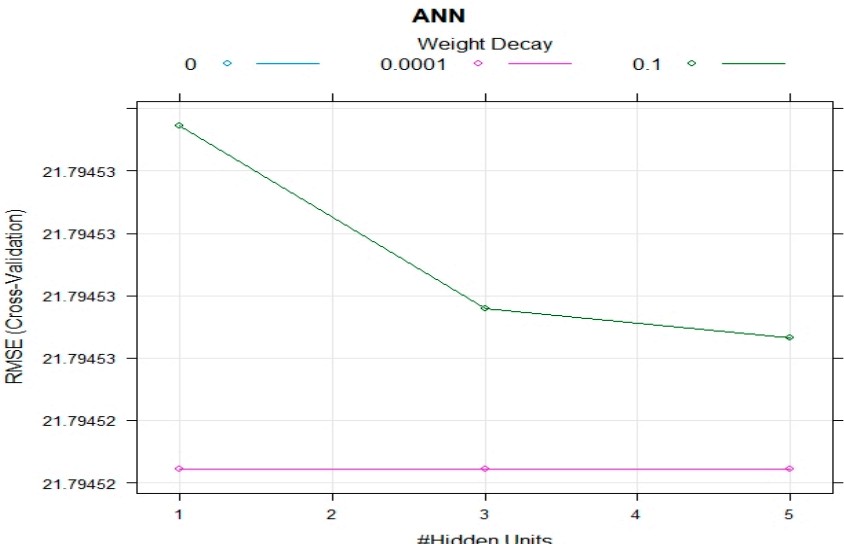

**Figure 9.** ANN RMSE result.

According to the Phillips–Perron test result of h = 1, the initial difference may be adequate to keep the monthly and daily temperature energy data stationary. Following the initial contrast, Figure 10 illustrates the stationary time series data for both time and monthly smart farm energy. The data oscillate around zero, showing that they are stationaries. This conclusion suggests the initial difference is sufficient and that additional data processing is not required. They found energy data substantial autocorrelations at delays multiples of 12, showing seasonality action every 12 months. At lower delays, however, there is potentially significant autocorrelation.

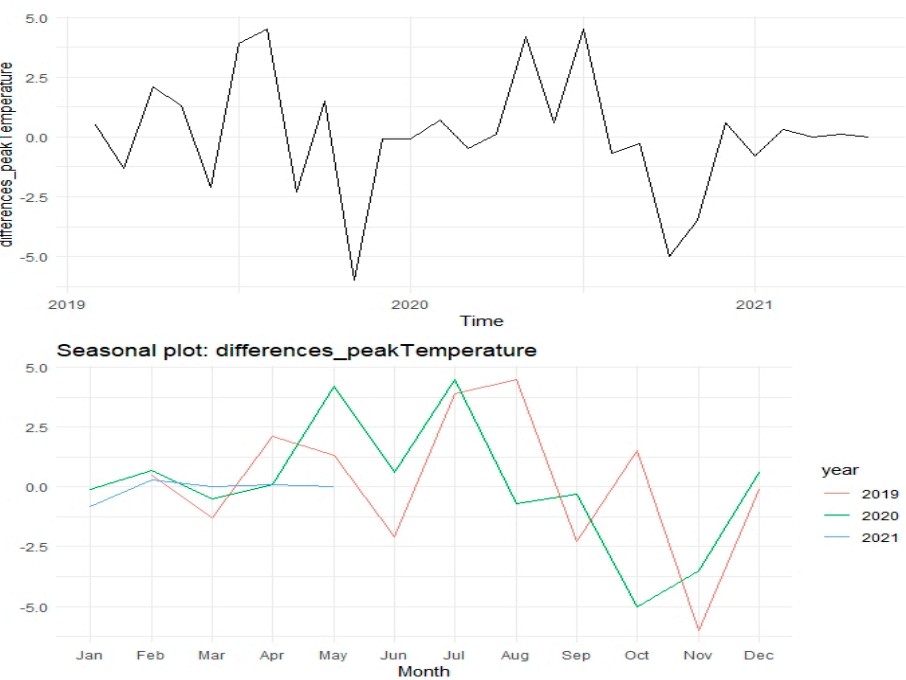

**Figure 10.** Autocorrelation seasonal peak temperature 2019 to 2021.

Figure 11, the ARIMA (0, 1, 1) with seasonal auto-regression at a lag of 12 and moving average at a lag of 12 has PACF decaying after the fourth lag and ACF decaying after the second lag. All potential combinations were tried to find the model with the minimum RMSE. Furthermore, the model produced satisfactory statistically meaningful findings for daily temperature energy figure after the second lag of ACF with a short lag decay for

PACF. As a result, ARIMA was deemed to be the final model (0, 1, 0). To find the optimum model and simplify the modeling process, all feasible combinations were tried again. An AR lag and an MA lag of 12 describe monthly temperature energy's ARIMA (0, 1, 1). With an RMSE of 4.28 and R2 = 69 percent, the constructed ARIMA model may be represented.

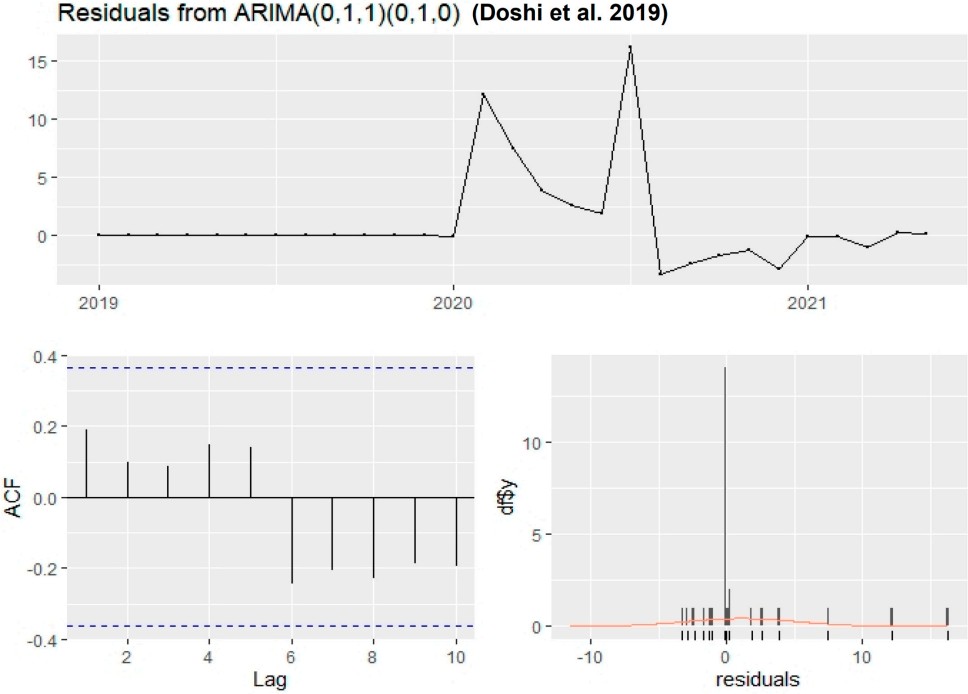

**Figure 11.** ARIMA final residuals (p, d, q), (0, 1, 1) [12].

ARIMA model with smart farm monthly and daily simulations temperature for the coming year and month. Although the smart farm simulation crop has more significant volatility, using this model and the ARIMA model is nearly identical. August had the maximum temperature, with an average of around 32 °C, just like the smart farm statistics. December was predicted to have the lowest monthly solar radiation (−1 to 6 °C), corresponding to the coldest month in actual temperature measurements. The overall trend of monthly temperature, on the other hand, has risen over time. Figure 11 shows that the average temperature in 2019 increased by around 2.5 percent in August, whereas the average temperature decreased by almost ten percent in 2020 August.

It is critical to select appropriate hyper-parameters in a prediction model to recognize and handle the prediction algorithms' error values. The CARET package in the R language contains a grid search function to find the best parameters for a prediction model. The LR model is linear with no tuning parameters; the only linear equation is used. On the other hand, other regression methods include a separate tuning parameter for each model.

CART is a decision tree that uses the shortest tree possible to contain the training data. The complexity parameter, the stopping criteria for CART models, is the tuning parameter considered for CART. 0.0001, 0.001, 0.01, 0.1, 0.2, 0.3 were the results of a grid search for complexity parameter values. The complexity parameter's excellent value was discovered to be 0.2, which was utilized to develop the model. The graph shows that RMSE remains constant for 0.2 and 0.3; hence, 0.2 was chosen as the best value.

Aside from the predictors, the SVM-radial basis kernel two tuning parameters are required for the model: sigma and cost. Cost regulates the penalty of misclassification and sigma is the radial basis kernel parameter. The sigma parameter was set to 0.1, 0.35, and 0.4, and the cost was found to be between 0.5, 0.9, 1.0, 1.1, 1.2. The grid search presented in Figure 12 shows the optimum SVM tuning parameters sigma (0.1) and cost (12).

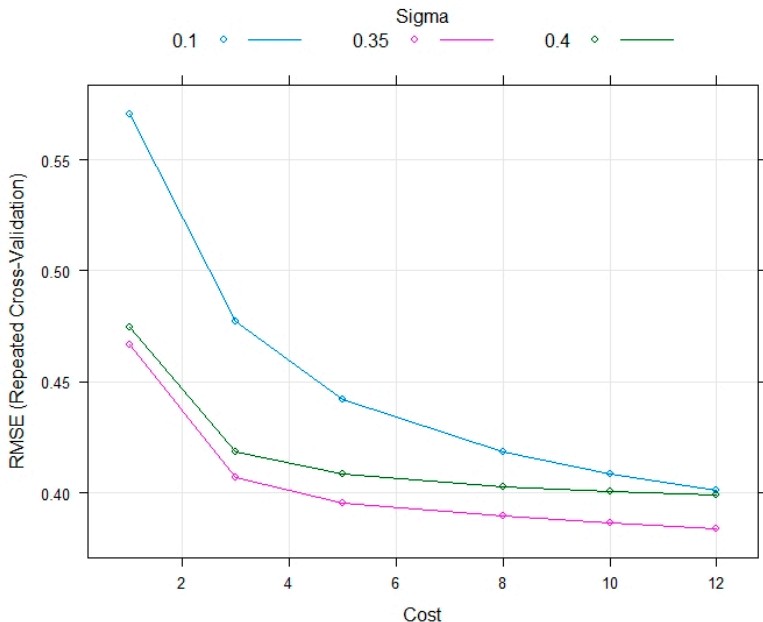

**Figure 12.** SVM result.

    kNN has many neighbor parameters, k, that need to be optimized. Because the kNN model predicts values by averaging neighbors, the number of neighbors required for the prediction is critical. The ideal value for k tested with k = 5, 7 and 9 was determined to be 1.98; in Figure 13, grid search for kNN spanned from 1 to 3, and the value for K was 9 which is the best RMSE value. The RMSE value is tiny when k is at 1.98.

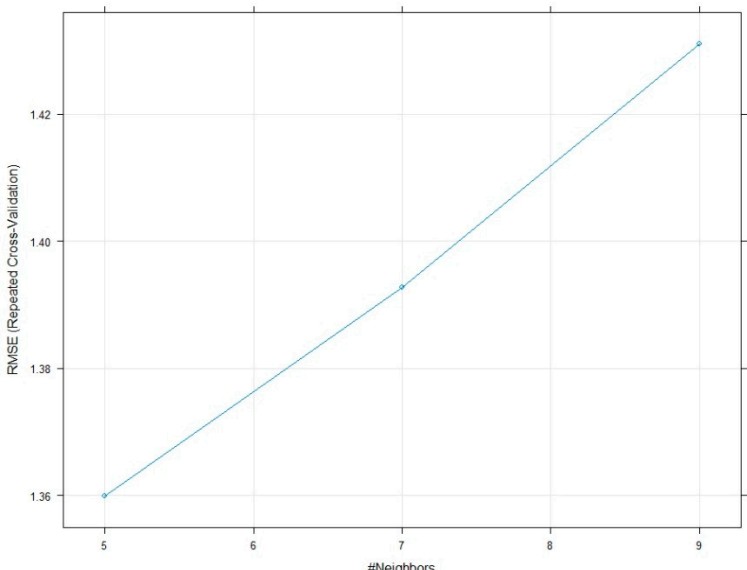

**Figure 13.** kNN result.

    RF, GBM, and XGB are tree-based models. Multiple regression trees are used in the performance of RF, GBM, and XGB. Each built each max tree using a random sample of chosen predictors. They were accomplished by improving the forecast and the de-correlation of the trees. RF, GBM, and XGB seek the best number of trees, but the number of predictors is random. As shown in Figure 14, the optimum number of predictors is 12, implying that RF considers all predictors and that the RMSE value remains constant after 500 trees. In Figure 15, when the depths of trees in GBM are 3, 6, and 9, the optimum number of predictors is 0.5 because GBM considers all predictors and the RMSE value remains constant after 2000 trees.

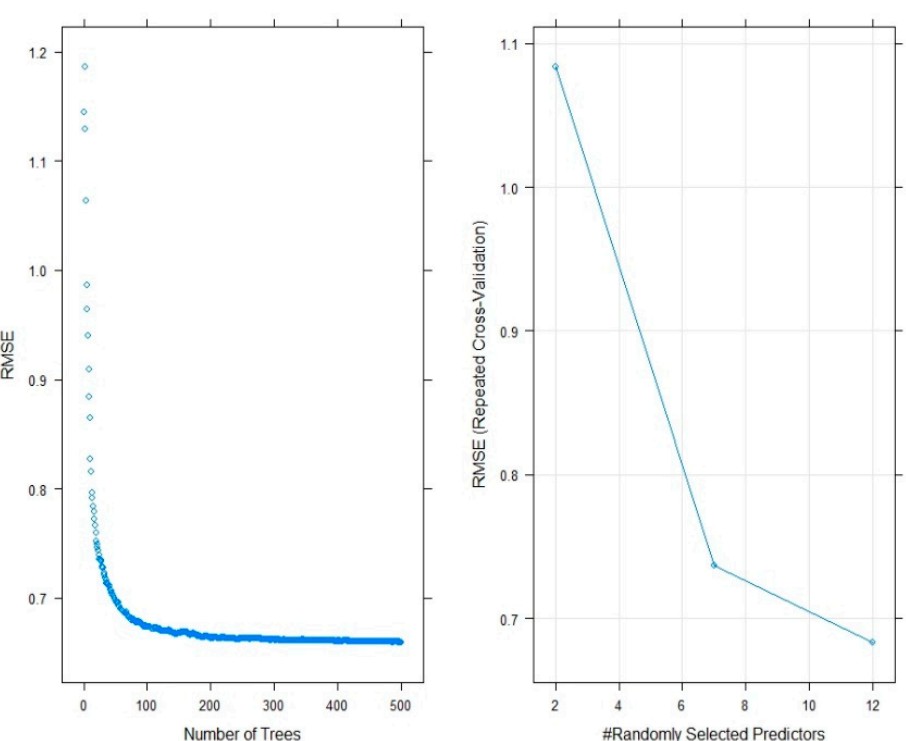

**Figure 14.** RF result.

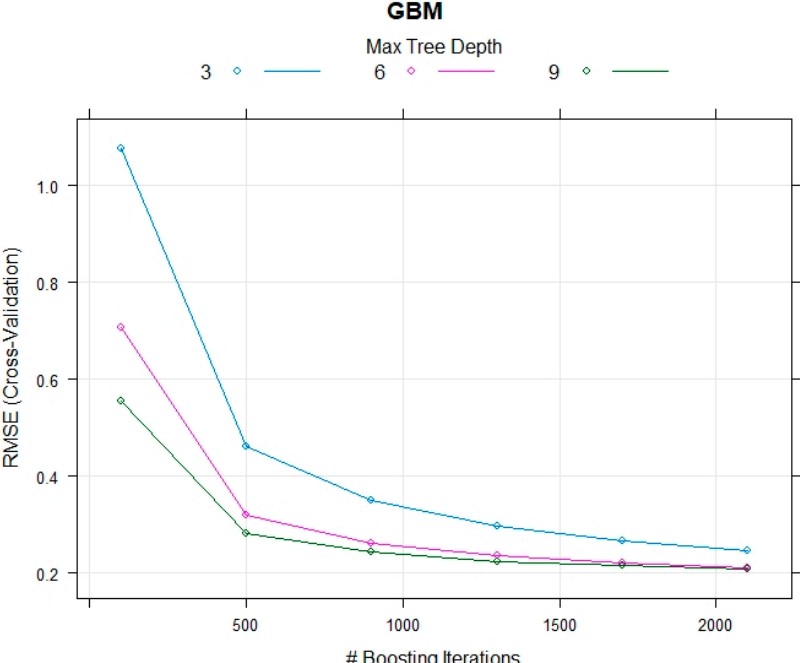

**Figure 15.** GBM result.

The ten-fold cross-validation approach was used to avoid over-fitting for the training XGBoost model. As a result, the best XGBoost model was found to have the following encouraging repetition parameters: maximizing repetitions = 140 and maximum tree depth = 3. Figure 16 depicts the XGB's performance when all predictors are taken into account, and the RMSE value remains constant after 140 trees. The optimal number of predictors is 2.5, as shown in Table 4.

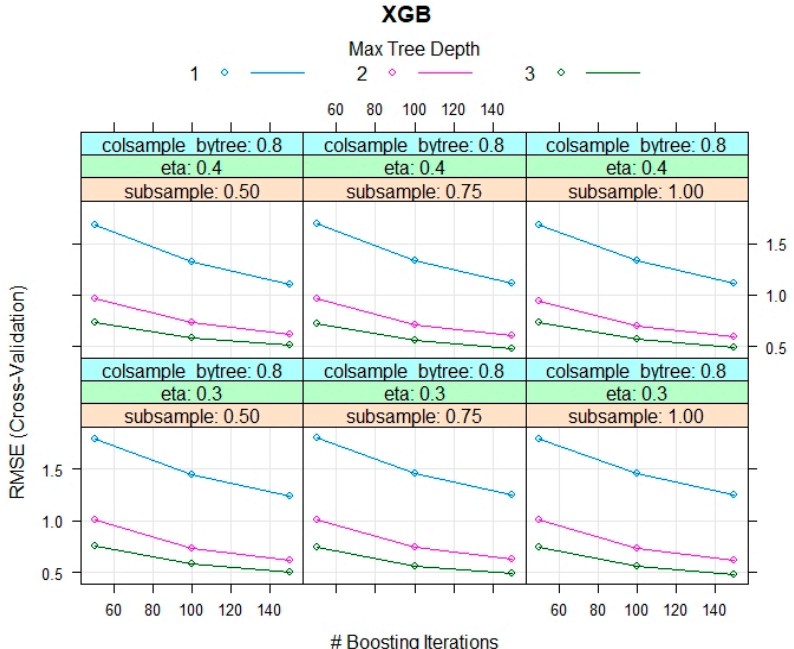

**Figure 16.** XGB result.

**Table 4.** The ML and ARIMA model performance.

| Models | Training | | | | Testing | | | |
|---|---|---|---|---|---|---|---|---|
| | **RMSE** | $R^2$ | **MAE** | **MAPE** | **RMSE** | $R^2$ | **MAE** | **MAPE** |
| ARIMA | 4.24 | 0.96 | 1.98 | 132.0 | 4.01 | 0.97 | 1.86 | 124.3 |
| LR | 2.09 | 0.78 | 1.73 | 26.5 | 2.67 | 0.84 | 1.37 | 25.46 |
| ANN | 21.79 | 0.18 | 21.23 | 95.2 | 21.28 | 0.18 | 21.24 | 95.26 |
| kNN | 1.97 | 0.83 | 1.28 | 26.7 | 1.84 | 0.94 | 1.48 | 26.8 |
| SVM | 0.37 | 0.99 | 0.25 | 27.3 | 1.05 | 0.95 | 1.79 | 26.81 |
| GBM | 0.65 | 0.98 | 0.37 | 27.4 | 1.08 | 0.97 | 1.68 | 26 |
| RF | 0.34 | 0.99 | 0.12 | 27.3 | 1.01 | 0.93 | 0.62 | 26.63 |
| XGB | 0.70 | 0.99 | 0.48 | 25.8 | 1.03 | 0.96 | 1.06 | 25.3 |

## 8. Result and Discussion

This study focused on predicting peak energy in the data analytics-enabled smart farm using environmental energy factors.

To extract all the additional variables, feature engineering was done. The study analysis was performed to understand the patterns in the data better. The models were built with their ideal tuning parameter settings explanation, shown in Figures 9 and 12–16. This work contributes to the already existing growing field of research in various ways. This research reveals that linear regression, a component of rule-based learning, is a sophisticated empirical modeling approach that may help classical learning algorithms predict more accurately. These findings could provide a new alternative for researchers to predict energy in a smart farm industry. Furthermore, it helps to enhance the library of empirical modeling algorithms that focus on forecasting smart farm energy. Moreover, this article can provide an effective and viable way for smart farm managers or greenhouse farmers who can better manage the problem of agricultural energy economically and environmentally. Therefore, we hope that the recommended ML method will help improve the smart farm's energy use or their energy policies in various fields related to agricultural energy.

Comparative assessments highlight many elements that best perform the performance of ML algorithms. The most significant input characteristics are optimization techniques, configuration settings, and evaluation measures. Temperature forecasting systems have smart farm energy factors as input parameters.

Smart farm energy temperature maximum, lowest, and mean values are consistent factors across all research. Maximum, minimum, and average time (hour, day, or month), environmental parameters are just a few examples. In reality, these model inputs are used exclusively in the paper. Because prediction accuracy depends on time, the temperature during validation, and other variables, it is best to conclude the validity of estimations based on accuracy measures (RMSE, MSE, MAE, R-squared).

The ML models used in this study are in Table 4. Since R-Squared is measured and defines the relationship between the predicted values, this value should be close to 1 and their ideal tuning parameters, as stated in the model construction section, utilizing 10-fold cross-validation. After training each model using the training set, each model is tested using the testing dataset. Considering all the ML models in terms of R-Squared, we can see that the Table 4 R-Squared value of all the models is greater than 0.9454. The average R-square value for the kNN is 0.83 and LR is 0.78, then the best R-square value ARIMA, SVM, GBM, RF, and XGB and low-value ANN, respectively. These show Figures 9 and 12–16 methods that accurately reflect the hours of the day on average smart farm energy. As a result, the average MAE, RMSR, MAPE value for the GBM, SVM, RF, and XGB models, respectively. RMSE models provide information that is always positive that reveals short-term performance. Table 4 should show RMSE best value is close to zero.

Table 4 shows the performance of the ML and ARIMA models in both the training and testing datasets for smart farm. The best model has the highest $R^2$ values and the lowest RMSE, MAE, and MAPE. In the training set, RF and SVM gave the most significant results. However, RF outperformed SVM marginally in the testing set. Because each regression model's final performance criteria are the testing set, the RF model outperforms the other regression models in this case. ANN, ARIMA, LR, and kNN have the worst performance compared to other prediction models. Because RF of many GBM and XGB, it performs better than GBM and XGB. This shows that the group strategies used for the RF model work better than the SVM.

The variable group contributed the most to the prediction accuracy of ML-based models for energy. It suggests the importance of variable during hourly accumulated light estimation on the ANN model. However, the variable outside temperature contributed the most for RF, SVM, GBM, kNN, and XGB models during hourly accumulated light. The most important predictors for the accuracy of most models for outdoor temperatures shown in Figure 17 are the heating temperature and the dew point. Figure 17 shows the relative variable importance for the smart farm and energy temperature program datasets using ANN, kNN, SVM, GBM, XGB, and RF. The relative variable importance of the RF model has measured the model's usage of a linear combination of terms used by the algorithm. In the SVM model, calculate the variable importance using the residual sum of squares. GBM and XGB track surrogate splits in the tree development process, and main divides decide a variable contribution to prediction. Consequently, the relevance of a variable in the GBM model is determined by its contribution to each split. In XGB each new statistic is used to determine the suitability of the variable that is selected for the neighboring variable.

Variable significance analysis can be useful for evaluating the most significant components in each model. This analysis will be useful to detect the effect of the considered variables in the developed models and study the model more thoroughly. As shown in Figure 17, the most influential variable for the smart farm dataset is the outside temperature, dew point, and heating temperature or humidity temperature, all of the established prediction models rate it as one of the top three most influential factors prediction. Outside temperature, humidity, dew point, outside solar temperature, and heating temperature variables are also the most significant variables with smart farm energy program data and regression models consistently ranked these variables in the Top 3 positions as shown in Figure 8. These analyses show that energy data variables such as outside temperature and hourly solar radiation are the most influential variables in predicting smart farm sharing demand.

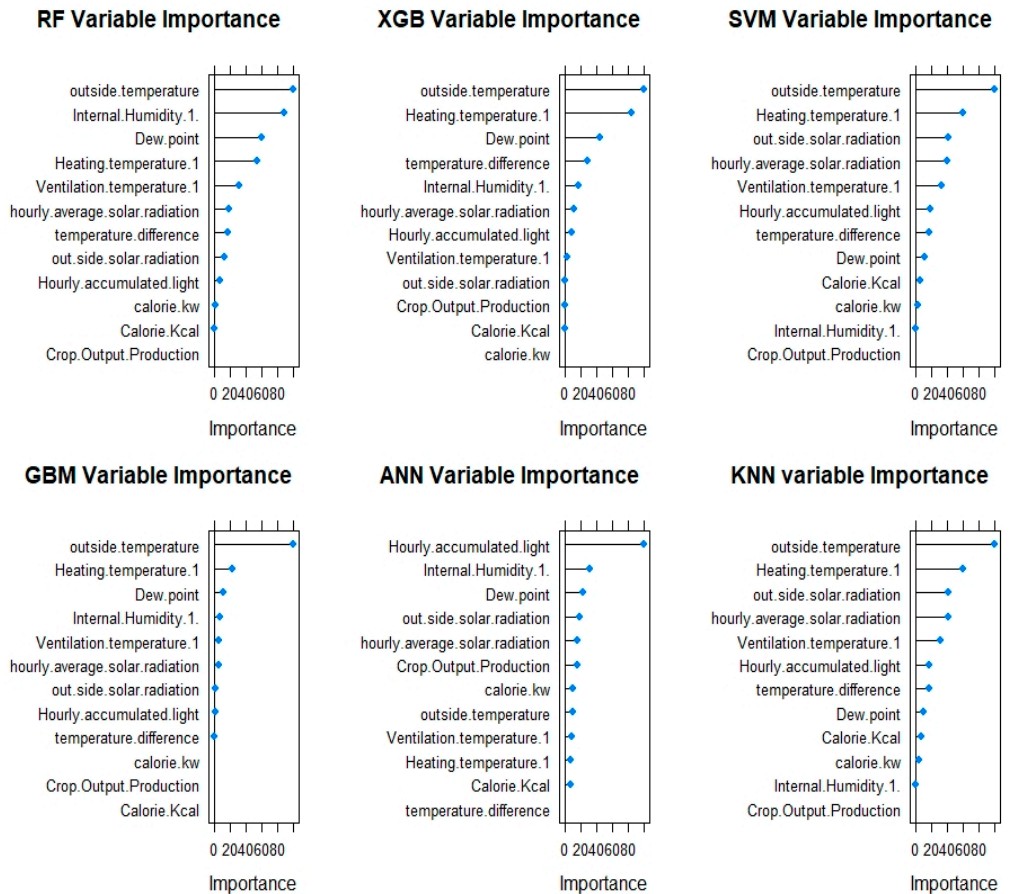

**Figure 17.** Variable importance.

Finally, shown in Table 4, a comparative test was conducted using the ML model, time series, and LR model, the RMSE, MAPE, MAE, and R square value of the selected models for crop growth using energy temperature in the smart farm. We have selected the most suitable model from the models listed in the tables and that has the lowest forecast error when comparing the predicted data using the correct test package. Further statistical analysis was carried out and the RMSE value is shown in Table 4. From the analysis above, the machine learning algorithm is the most suitable model for the data sets considered using the smart farm temperature they generate.

According to the findings, RF consistently produces better results than other methods. This model's flexibility allows it to perform better regardless of known or unknown choice rules. A rule-based RF model is used in this study. It implies that the model learns a variety of complex attributes about the temperature of smart farms. However, over-fitting the model can be a real issue when creating any regression model. In any case, model over-fitting can be a serious issue when developing a regression model. As a result, it is critical to feed the model with pre-processed data

## 9. Conclusions

This study focuses on predicting the peak temperature energy. The results show RF algorithm improves the R-squared, MAPE, RMSE, and MAE compared to ANN, SVM, GBM, XGB, kNN, and ARIMA models in the smart farm data. This shows that the RF model can predict temperature energy. The purpose of the variable significance analysis was to identify the variables' hidden relationships. Internal humidity and dew point, or outside temperature, were the essential variables in all the models for predicting smart farm energy. The results show that RF models effectively identify significant inputs related to energy usage for crop production, explaining the relationship between energy and internal

temperature change in smart farms. Our future research will focus on finding the worst and best predicted days using variable-temperature data in the smart farm. Discover the best crop growth day using ML algorithm models analysis using the parameters of various data measurement techniques and deep learning with big data.

**Author Contributions:** S.V. performed data curation and data pre-processing, formula analysis, methodology, validation and writing of the original manuscript; J.L.; methodology, Y.C. contributed to investigation, supervision, editing the manuscript, supervised and endorsed the study; H.K. supervised the study, investigation. All authors have read and agreed to the published version of the manuscript.

**Funding:** This research has been funded by the MSIT (Ministry of Science and ICT), Korea, under the Grand Information Technology Research Center support program (IITP-2021-2020-0-01489) supervised by the Institute for Information & communications Technology Planning & Evaluation (IITP). This work was supported by Korea Institute of Planning and Evaluation for Technology in Food, Agriculture and Forestry (IPET) through Smart Farm Innovation Technology Development Program, funded by Ministry of Agriculture, Food and Rural Affairs (MAFRA) and Rural Development Administration (RDA) and Ministry of Science and ICT (MSIT) (421028-3). This research was supported by Basic Science Research Program through the National Research Foundation of Korea (NRF) funded by the Ministry of Education (No. 2020R1A6A1A12047945. The Korea Institute of Energy Technology Evaluation and Planning (KETEP) grant funded by the Korean government (MOTIE) (20202020900060) have been sponsored this work. Interdisciplinary Program in IT-Bio Convergence System, 21Sunchon National University, 255, Jungang-ro, Suncheon-si, Jeollanam-do 57922, Korea.

**Conflicts of Interest:** The authors declare no conflict of interest.

## Abbreviations

| Abbreviation | Meaning |
|---|---|
| ML | Machine Learning |
| ANN | Artificial Neural Network |
| GBM | Gradient Boosting Machine |
| XGB | eXtreme Gradient Boosting |
| kNN | k-Nearest Neighbors |
| RF | Random Forest |
| SVM | Support Vector Machine |
| ARIMA | Autoregressive integrated moving average |
| LR | Linear Regression |
| WSN | Wireless sensor network |
| ARIMA (p, d, q) | p is the number of autoregressive terms, d is the number of non-seasonal differences needed for stationarity, and q is the number of lagged forecast errors in the prediction equation. |
| ACF | Auto Correlation Function |
| DTs | Decision Trees |
| kW | Kilowatt-hour |
| CARET (R package) | Classification and Regression Training |
| CART (decision tree) | Classification and Regression Trees |
| RMSE | Root Mean Square Error |
| MSE | Mean Square Error |
| MAE | Mean Average Error |
| $R^2$ | R-squared |

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
