# Peer review of "A Machine Learning Based Model for Energy Usage Peak Prediction in Smart Farms"

_electronics, doi:10.3390/electronics11020218_

Round 1

Reviewer 1 Report

I believe that the article is interesting, but it has many aspects to improve that prevent me from recommending its publication in its current state.
1- The motivation of the paper could be improved by including a clearer organizational structure: context, a problem addressed, research questions to be solved, hypotheses, scientific method to validate the hypotheses, etc.

2- The related work seems to me to be clearly scarce and unjustified. I believe that the authors should carry out a much more extensive SLR, detailing the methodology used (search criteria, key terms, inclusion and exclusion criteria, etc.). This would make it clear what similar approaches exist to solve the same open problem faced.

3. I strongly recommend, after related work give a separate section for problem statement for better understanding. In methodology, the author has given an explanation only state of the art machine learning algorithms but the title mentioned: “Novel Machine Learning Based Model for Energy Usage Peak Prediction in Smart Farms”. I didn’t find out any novel methodology in this paper.

4- The validation is not sufficiently justified: what real situation does the scenario simulate? is it realistic? is it enough to fully validate the proposal? why?

5. In conclusion author mentioned RF-based model has given better results, I strongly recommend comparing the recent papers

Author Response

Respected Reviewer 1, I have attached reviewer report 1. Please see the attachment.

Reviewer 2 Report

Comment:
1. It is recommended to explain all abbreviations in the paper such as WSN, LR, XGB, SVM, RF, GBM, KNN, ARIMA, and ANN.
2. Revised Figure 1, Figure 5, Figure 6 with a good resolution.
3. Check your flowchart in Figure 2. The decision marks should be Yes and No.
4. Consistent in writing kNN or KNN
5. It is suggested to change Figure 7 to (a) and (b), because it has two pictures.
6. The total number of Figures is 20, in my opinion too many, the author needs to reduce it and add a more in-depth discussion of each figure.
7. The author state 
"This study shows the smart farm crop energy rainy day and not rainy day data from 342
January 2019 to May 2021 (29 months) in Figure 5." It is suggested to do another experiment with another dataset to prove that your proposed method is robust.
8. The author state that RF is the best effective model for forecasting yearly energy use in this study. Many previous research studies have stated that RF is the best model.
Cite this related research paper:
R. C. Chen, C. Dewi, S. W. Huang, and R. E. Caraka, “Selecting critical features for data classification based on machine learning methods,” Journal of Big Data, vol. 7, no. 52, pp. 1–26, 2020. Based on that fact, explain your major contribution to the paper.
9. Compare your research results with previous research studies, authors can also use public datasets as benchmarks.

Author Response

Respected Reviewer 2, I have attached reviewer report 2. Please see the attachment.

Round 2

Reviewer 1 Report

The author has incorporated necessary comments, now the article looks fine to publish. 

Reviewer 2 Report

Paper can be accepted now. The revised version is good enough and clear.